# The Role of EMT-Related lncRNAs in Ovarian Cancer

**DOI:** 10.3390/ijms241210079

**Published:** 2023-06-13

**Authors:** Dimitra Ioanna Lampropoulou, Marios Papadimitriou, Christos Papadimitriou, Dimitrios Filippou, Georgia Kourlaba, Gerasimos Aravantinos, Maria Gazouli

**Affiliations:** 1ECONCARE LP, Health Research & Consulting, 11528 Athens, Greece; d_lambropoulou@yahoo.gr; 2Myeloma Division, Sylvester Comprehensive Cancer Center, University of Miami, Miami, FL 33136, USA; mxp2050@miami.edu; 3Second Department of Surgery, Aretaieion Hospital, Medical School, National and Kapodistrian University of Athens, 11527 Athens, Greece; 4Department of Anatomy and Surgical Anatomy, Medical School, National and Kapodistrian University of Athens, 11527 Athens, Greece; d_filippou@hotmail.com; 5National Organization for Medicines (EOF), 15562 Athens, Greece; 6Department of Nursing, University of Peloponnese, 22100 Tripoli, Greece; kurlaba@gmail.com; 7Euroclinic, Athanasiadou 7-9, 11521 Athens, Greece; garavantinos@yahoo.gr; 8Department of Basic Medical Sciences, Laboratory of Biology, Medical School, National and Kapodistrian University of Athens, 11527 Athens, Greece; mgazouli@med.uoa.gr

**Keywords:** lncRNAs, ovarian cancer, EMT, biomarkers

## Abstract

Ovarian cancer (OC) is one of the deadliest cancers worldwide; late diagnosis and drug resistance are two major factors often responsible for high morbidity and treatment failure. Epithelial-to-mesenchymal transition (EMT) is a dynamic process that has been closely linked with cancer. Long non-coding RNAs (lncRNAs) have been also associated with several cancer-related mechanisms, including EMT. We conducted a literature search in the PubMed database in order to sum up and discuss the role of lncRNAs in regulating OC-related EMT and their underlying mechanisms. Seventy (70) original research articles were identified, as of 23 April 2023. Our review concluded that the dysregulation of lncRNAs is highly associated with EMT-mediated OC progression. A comprehensive understanding of lncRNAs’ mechanisms in OC will help in identifying novel and sensitive biomarkers and therapeutic targets for this malignancy.

## 1. Introduction

Ovarian cancer (OC) is the fifth highest cause of cancer-related death among women and the most lethal gynecological cancer [1]. Treatment success is arguably related to early diagnosis since 5-year survival surpasses 90% in women diagnosed with stage I OC [2]. However, ineffective screening strategies and vague symptomatology lead to delayed diagnosis at late stages with extended intraperitoneal dissemination of the disease. Moreover, despite radical tumor debulking surgical techniques and new targeted therapeutic approaches, such as antiangiogenics and poly (ADP-ribose) polymerase (PARP) inhibitors, the prognosis of OC is still poor, with the overall 5-year relative survival rate ranging between 30–40%, globally [3]. In general, OC encompasses three subtypes: stromal, germ, and epithelial cell tumors, with the latter accounting for approximately 90% of all cases [4]. The current standard of care in OC includes a combination of aggressive surgical cytoreduction and cytotoxic chemotherapy (predominantly platinum and paclitaxel). Interestingly, the pre-operative frailty assessment of patients undergoing non-emergent surgery has been proposed for the prediction of surgery-related complications, disease-free survival (DFS), and overall survival (OS) [5]. However, chemoresistance remains one of the largest pitfalls of treatment.

Epithelial-to-mesenchymal transition (EMT) refers to a dynamic process that occurs both under physiological conditions, i.e., during embryogenesis and tissue regeneration (ΕΜΤ- type 1 and 2, respectively), and pathological processes, such as cancer (EMT-type 3) [6,7]. EMT is a continuous and reverse process [ranging between EMT and mesenchymal-to-epithelial transition (MET)], during which epithelial cells can convert properties along the E to M spectrum, shifting from epithelial to mesenchymal phenotypes. During this shifting process, several molecular, morphological, and functional changes take place, including cellular polarity, intercellular and cell-matrix adhesion, and cytoskeleton remodeling [8]. In terms of cell dynamics, cancer cells exist in intermediate states of EMT, expressing both epithelial and mesenchymal genes; those hybrid cells exhibit improved survival, migratory and colonizing potential [9]. A complex mosaic of factors plays a crucial role in the regulation of EMT. This network includes transcriptional and posttranslational control and epigenetic modifications as well as non-coding RNA-mediated regulation [6]. During the past decade, EMT has become a field of extensive scientific interest in cancer progression and treatment resistance. 

The continual scientific quest to decode and comprehend the function of the human genome led to the remarkable discovery that only 2% of transcribed DNA encodes for proteins [10]. Recent advances in sequencing technologies have helped significantly in interpreting the role of non-coding DNA and the corresponding non-coding RNAs (ncRNAs). NcRNA is an umbrella term under which fall several subtypes that play significant roles in cellular functions. NcRNAs are typically divided by their length in long non-coding RNAs (lncRNAs), which represent transcripts more than 200 bp long, and small ncRNAs [such as microRNAs (miRNAs, miRs), small nuclear RNAs (snRNAs), small inhibitory RNAs (siRNAs) and Piwi-interacting RNAs (piRNAs], which are shorter than 200 bp [11]. A plethora of studies have supported the biological relevance of lncRNAs and underlined their central functions in both physiological and pathological conditions, such as cell differentiation and proliferation, immune response, tumor genesis, and tumor invasion and migration [12]. Accordingly, an increasing number of studies highlight their participation in cancer progression and metastasis via regulating EMT. Interestingly, despite usually being divided into EMT inducers and EMT inhibitors, many display controversial and contradictory roles depending on the cancer type, further pinpointing the complex makeup of tumor cells (Figure 1). 

The combination of serum cancer antigen 125 (CA125) levels with transvaginal ultrasonography (TVUS) remains the traditional approach to the initial evaluation of suspected OC cases. However, CA125 has questionable specificity due to its increased rate of false positive results [13]. Similarly, TVUS cannot accurately differentiate between benign and malignant masses [14]. Moreover, although it has been suggested that a panel of biomarkers may be more effective for personalized treatment [15], CA125 remains the only widely used OC biomarker in clinical practice. Given the above, the clinical need for the identification of novel biomarkers for early detection, prognosis, and response when treating OC is more than essential. 

In this review, we investigated current literature for EMT-related key regulatory lncRNAs involved in OC in an attempt to summarize existing information and provide a network of lncRNAs that could serve as new diagnostic and prognostic biomarkers, as well as therapeutic targets. 

## 2. Brief Overview of EMT in Oncology

Tumor progression has been significantly linked with EMT, and therefore it has captured increasing interest for researchers. Indeed, the association of EMT with cancer has been reported since the early 80s [16]. EMT refers to an escape mechanism that takes place in cancer cells and is associated with tumor development, invasion, and metastasis to secondary sites [17]. This process includes the transdifferentiation of epithelial cells to a mesenchymal phenotype which exerts invasive and migratory properties [18]. During the orchestration of EMT, three biological phenomena may occur simultaneously. The first one involves the disruption of cell-to-cell contact and epithelial tissue polarity. Second, the remodeling of the cytoskeleton leads to the restructuring of epithelial cells to fibroblast-like cells, and third, several changes in epithelial and mesenchymal markers occur. These changes concern alterations to EMT-related proteins, including a decrease in E-cadherin expression (an epithelial-specific factor) and an overexpression of N-cadherin and vimentin (mesenchymal-specific factors). Moreover, the switch in gene expression from epithelial to mesenchymal phenotype is also triggered by (a) EMT-inducible transcription factors (*ZEB1*, *ZEB2*, *Snail*, *Slug*, and *Twist*), (b) non-coding RNAs, (c) epigenetic modifications and posttranslational regulation, and (d) alternative splicing events [19,20,21,22]. Over the years, multiple triggering factors of EMT have been identified, including (i) microenvironment signals (such as hypoxia and oxidative stress) [23], (ii) growth factors [epidermal growth factor (EGF), and fibroblast growth factor (FGF)] [24,25], and cytokines secreted by the tumor microenvironment [tumor necrosis factor-alpha (TNF-α), IL-8, IL-60] [26,27,28], (iii) immune responses (such as elevated expression of immune checkpoints, namely PD1, PD-L1, LAG3, and CTLA4) [29], and (iv) regulation of drug- and radio-resistance-related genes [23]. Indeed, there is an intriguing relationship between EMT and the immune system. There is evidence that the cells of the innate and adaptive immune system (i.e., macrophages, natural killers, and Tregs) promote EMT [30,31], whereas mesenchymal-like tumor cells possess immunosuppressing functions [32]. More importantly, surgical debulking of OC has been linked to significant reductions in circulating Tregs and increases in CD8+ T-cells, supporting the idea that upfront cytoreduction has a valuable systemic effect by improving immunological state [33]. Furthermore, tumor-infiltrating lymphocytes (TILs) have been also associated with optimal primary surgical cytoreduction [34]. A recent comprehensive review by Huang et al. thoroughly describes all the molecular mechanisms of EMT involved in tumor progression and metastasis [6]. 

## 3. EMT in Ovarian Cancer 

Wu et al. first reported that increased levels of E-cadherin can induce cell-to-cell cohesion, preventing cellular migration in human OC cells [35]. This study highlighted the importance of EMT in tumor cell shedding into the peritoneum and intra-abdominal metastases. Indeed, as opposed to other epithelial malignancies that disseminate via the canonical invasion–metastasis cascade mechanism (which includes the hematogenic or lymphatic route) [36], epithelial ovarian cancer (EOC) metastasizes predominantly via a transcoleomic route of spread. Briefly, this route involves the peritoneal cavity with direct cancer cell shedding (both as single cells and as cell aggregates) from the primary tumor into the peritoneal fluid. Subsequently, their adherence to intraperitoneal tissues leads to their anchoring into the sub-mesothelial matrix, where they generate secondary lesions [37,38]. This unique peritoneal microenvironment exhibits an exceptional phenotypic plasticity, promoting epithelial-to-mesenchymal and mesenchymal-to-epithelial transitions. More specifically, OC cells exist in the peritoneal cavity in a heterogenous variety of phenotypes along the EMT spectrum (epithelial, hybrid, and mesenchymal), with the ability to dynamically shift from one state to another, responding to micro-environmental stimuli. Epithelial-type cell aggregates have been associated with radio- and chemo-resistance as well as resistance to apoptosis. Several bioactive soluble factors [lipid lysophosphatidic acid (LPA), growth factors, and matrix metalloproteinases] facilitate the generation of a soluble E-cadherin fragment which may trigger the disruption of intercellular junctions between cells with epithelial and hybrid phenotypes and promote cell exfoliation from the primary tumor. On the other hand, mesenchymal-type cell aggregates have been associated with metastasis via the invasion of peritoneal matrix. The latter can be also enhanced by the soluble E-cadherin fragment. Finally, the hybrid-type cells exhibit stemness properties with tumor initiation and growth potential. In mixed-cell aggregates (hybrid–epithelial or hybrid–mesenchymal), hybrid cells may differentiate to the adjacent cell phenotype and share their functions [39]. 

Several EMT regulators have been identified in OC and described in depth elsewhere [39]. In brief, besides non-coding RNAs, various components of the ascitic fluid, epigenetic changes, post transcriptional modifications, and biomechanical forces have been proposed to trigger EMT. For example, the role of LPA in EMT promotion has been widely established. Gil et al. reported that LPA favors the mesenchymal phenotype with enhanced invasion potential, via the shedding of an 80-kDa E-cadherin-soluble fragment in a urokinase plasminogen activator (uPA)-dependent manner [40]. EMT induction by LPA in OC has been also associated with (i) *SIRT1* downregulation, a known *ZEB1* inactivator and EMT suppressor [41] and (ii) *Slug/Snail2* upregulation via the Gαi2, Src and HIF1α signaling nexus [42]. In addition, LPA has been correlated with matrix metalloproteinase-9 (MMP-9) overexpression and MMP-9-associated E-cadherin ectodomain shedding, which results in the disruption of cell–cell junctions between cancer cells [43]. Following the dysfunction of junctional integrity, LPA further stimulates EMT via the β1-integrin-dependent activation of Wnt/β-catenin signaling pathway [44]. Finally, it has been suggested that MMP-2 activation by LPA leads to enhanced migratory and invasive potential of epithelial OC cells [45]. 

Besides LPA, growth factors [such as epidermal growth factor (EGF), *transforming growth factor beta* (TGF-β) and hepatocyte growth factor (HGF)] which are also present in the ovarian tumor microenvironment have been associated with EMT induction through (i) the promotion of the cleavage of Ecad fragments to abrogate cell–cell adhesion and (ii) the activation of complex signaling pathways [46,47,48]. Moreover, MMPs facilitate EMT by inducing the cleavage of the Ecad ectodomain and suppressing the tight junction protein zona occludens-1 (ZO-1). These processes lead to impaired intercellular cohesion and activation of the Wnt signaling pathway, resulting in enhanced mesenchymal properties and thus increased invasive potential in the tumor cells [49,50,51]. Simultaneously, EMT-related factors such as *Snail* and β-catenin have been reported to regulate MMP expression [52,53]. Another component of ascitic fluid, the Wnt5a protein has been also associated with EMT shifts [54]. Furthermore, EOC-associated malignant ascites are enriched in IL-6 and IL-8, which have been believed to play a pro-EMT role in OC [55,56]. Notably, increased expression of both these cytokines has been also associated with LPA via the transcriptional activation of IL gene promoters [57]. 

In addition to the above, several EMT-associated epigenetic modifications have been described over the years. For instance, epigenetic silencing of *CDH1* (the gene that encodes e-cadherin) can result from (i) hypermethylation via the 5′ CpG island [58] (ii) methylation via *ZEB1* through the recruitment of DNA methyltransferase 1 (DNMT1) to the *CDH1* promoter [59] and (iii) mono-, di and trimethylation of the *CDH1* promoter [60]. On the other hand, reactivation of *CDH1* has been linked with deacetylation of *Slug* [61]. Another example of EMT-related epigenetic change in OC involves the TGF-β-induced methylation of several genes such as (a) *CDH1*, *FGFBP1*, *SNAI3*, and *MMP9*, resulting in their downregulation; and (b) *MMP2*, *MMP3*, *ZEB2*, *TGFB2* and *SNAI1*, resulting in their upregulation [62]. 

Accordingly, various modifications have been reported to regulate EMT at the posttranscriptional level. Briefly, phosphorylation of SNAI1 by protein kinases leads to its degradation and thus increased E-cadherin expression, whereas inhibition of these enzymes results in decreased levels of E-cadherin and EMT promotion [63]. Moreover, Snail1 glycosylation prevents its degradation, thus inducing EMT [64]. The mediation of EMT in the posttranscriptional setting has been also linked with the impairment of critical molecules of the HIPPO signaling pathway (namely YAP/TAZ, MAT1/2, and LATS1/2) due to abnormal regulation of posttranscriptional modifications [65]. Finally, sumoylation of Smad-Interacting Protein 1 (SIP1) has been correlated with E-cadherin downregulation [66]. The following sections will describe EMT-related mechanisms in ovarian cancer initiation and progression, focusing on the role of long non-coding RNAs. 

## 4. Regulation of EMT by Long Non-Coding RNAs in Ovarian Cancer: Molecular Mechanisms 

LncRNAs represent a subtype of non-coding RNAs with distinct regulation properties in cancer biology, including cellular proliferation, invasion, and metastasis [8]. Although a growing number of studies have demonstrated their implications for protein-encoding genes, their role in terms of regulating EMT/MET in ovarian cancer has been relatively recent. Table 1 summarizes and Figure 2 illustrates the proposed molecular mechanisms of the oncogenic lncRNAs that have been implicated in OC-related EMT until now. 

Recently, lncRNA AC005224.4 was found to exhibit significant oncogenic properties, affecting EMT-associated markers in OC cell lines. The authors suggested that this lncRNA promotes OC cell proliferation, metastasis, and EMT via sponging miR-140-3p and thus increases *SNAI2* expression [67]. Plasmacytoma-variant translocation 1 (PVT1) enhanced the proliferative and migrative potential of OC cells through the modulation of the expression of connective tissue growth factor (CTGF), which in turn also promotes the EMT process [69]. Moreover, SNHG17 knockdown was found to repress EMT in OC cells via the regulation of miR-485p/AKT1 axis [71]. Another lncRNA, OIP5-AS1, was upregulated in OC. Downregulation of OIP-AS1 repressed EMT, migration, and invasion while facilitating apoptosis of OC cells. More specifically, this lncRNA was reported to function as a competing endogenous RNA (ceRNA) by sponging miR-92a to regulate integrin alpha 6 (ITGA6) [72]. Similar conclusions were reported in a second study; in the latter, the investigators supported the idea that OIP-AS1 acts as ceRNA for miR-137, resulting in *ZNF217* upregulation and EMT promotion [73]. HCG18 silencing suppressed proliferation, migration, and EMT in OC cells and reduced ki67 levels in vivo. This lncRNA also functions as a ceRNA for miR-29a/b, resulting in *TRAF4/5* downregulation, which subsequently activates the NF-κB signaling pathway [74]. Similarly, LINC00909 acts as a ceRNA for MRC2 mRNA by sponging miR-23-3p and thereby induces EMT in OC cells [82]. Liu et al. proposed that hypoxia-induced MIR210HG was positively correlated with poor prognosis and cancer progression. Further results of their work demonstrated that knockdown of MIR210HG suppressed EMT and angiogenesis under hypoxic conditions [75]. Another widely studied lncRNA in cancer is MALAT1. According to a recent work, MALAT1 functions as a ceRNA interfering with miR-22 and consequently increasing the expression of Snail in OC [76]. EMT promotion in OC cells by MALAT1 has been also associated with the regulation of additional EMT hallmarks, such as *ZEB2* [77] and *RBFOX2* [78]. Moreover, MALAT1 expression has been negatively correlated with 5-year OS and it has been suggested that it induces EMT via the PI3K/AKT pathway [79]. Similarly, increased expression of the oncogene H19 promotes the OC-related EMT phenotype via direct binding to miR-140-5p and activation of the PI3K/AKT pathway [104]. In addition, H19 has been reported to facilitate TGF-β-induced EMT by sponging miR-370 in OC cells [103], while SRA overexpression regulated EMT-related markers and the NOTCH pathway [81]. A work conducted by Xue et al. linked lncBCAS1-4_1 upregulation with EMT and vitamin D signaling. According to the researchers, lncBCAS1-4_1 could eradicate the antitumor function of the 1α,25(OH)2D3 due to ZEB1 upregulation [80]. The effect of LINC00922 on OC cell proliferation, invasion, migration, and EMT was investigated by Wang et al.; in brief, they reported that this lncRNA competitively binds with miR-361-3p, resulting in the upregulation of CLDN1 and activation of the Wnt/β-catenin signaling pathway [83]. Indeed, the enhancement of the Wnt/β-catenin signaling pathway in OC has been associated with more than one lncRNA. For example, alongside the regulation of epithelial and mesenchymal markers, SNHG8 silencing resulted in downregulation of EMT-associated stem factors such as *ALDH1*, *Nanog*, *SOX2*, *OCT4*, *CD44*, and *CD133* through the Wnt/β-catenin pathway [95]. In addition, HCP5 was reported to (a) serve as a sponge for miR-525-5p, preventing its further binding to polycomb repressive complex 1 (PRC1) and (b) promote the Wnt/β-catenin signaling pathway [96]. Moreover, Lou et al. found that ROR promoted EMT contribution to cell migration and invasion through the activation of the canonical Wnt/β-catenin signaling pathway [107]. It has also been suggested that this lncRNA acts as a sponge for the tumor-suppressive miR-145 and its target gene FLNB [106]. Finally, two more lncRNAs, HOXB-AS3 and CCAT2, exert their EMT-promoting functions through activation of the Wnt/β-catenin signaling pathway [110,119]. 

EMT-mediated OC progression has been also associated with a positive feedback loop of the lncRNA DSCR8 via the miR-98-5p/STAT3/HIF-1α nexus [92]. The activation of the STAT3 signaling pathway has been also attributed to the lncRNA ATB [102]. PAXIP1-AS1 promotes cell proliferation and EMT by sponging miR-6744-5p to regulate PCBP2 expression [84], whereas MAFG-AS1 was reported to facilitate EMT progress by upregulating the NFKB1-dependent IGF1 via miR-339-5p [86]. Another lncRNA, CTSLP8, was found to promote OC in vitro and in vivo by sponging miR-199a-5p. Mechanistically, the authors reported that lnc-CTSLP8 exerted its oncogenic effects through *CTSL1* upregulation [87]. EMT-related markers were also identified as regulated by HOTIPP overexpression via activation of the MEK/ERK pathway [91]. Moreover, nuclear-enriched abundant transcript 1 (NEAT1) was found to sponge miR-1321 and thus regulate the expression of the tight junction protein 3 (TJP3) (an important factor in cancer development), resulting in EMT promotion, invasion, and migration of OC cells [93]. LINC00858 has also been associated with OC proliferation, motility, and EMT by sponging miR-134-5p to upregulate *RAD18* [94]. The findings of another study demonstrate that E2F4as promoted tumor aggression by regulating EMT-associated mechanisms, thereby predicting patients’ prognosis [89]. Additionally, the EMT phenotype in OC cells has been found to be mediated by (a) MIAT by sponging miR-150-5p [99], (b) NORAD by regulating miR-199a-3p [100], and (c) LINC00963 by targeting miR-378g [101]. PTAL (also known as AC004988.1) has been also associated with facilitating EMT via the miR-101-FN1 axis [105]. Furthermore, FLVCR1-AS1 was found to bind directly to miR-513, thus downregulating its expression. In addition, the authors reported that miR-513 underexpression might be linked with *YAP1* upregulation [109]. Similarly, according to Yan et al., DQ786243 could directly bind to miR-506 and thus increase cAMP responsive element binding protein 1 (CREB1) expression [115]. Another study, using bioinformatics analysis, noted that lncRNA PTAR expression was positively correlated with *ZEB1* expression in OC; the same work also suggested that PTAR enhanced EMT by binding to miR-101-3p [116]. In addition, Linag et al. proposed that lncRNA PTAF regulated EMT in OC through the miR-25-SNAI2 axis [118]. 

The association between lncRNA expression levels, EMT, and OS in OC patients has been also investigated in several studies. For example, LINC01969 was linked with OC progression and worse OS by regulating the miR-144-5p/*LARP1* axis as a ceRNA [88]. Similarly, LINC01094 overexpression has been associated with higher FIGO stage, lymph node (LN) metastasis, and shorter OS in patients with OC. Conversely, LINC01094 silencing retarded OC cell proliferation, migration, invasion, and EMT by targeting miR-577 and activating the Wnt/β-catenin signaling pathway [90]. Liu et al. suggested that LINC01215 promotes EMT-related LN metastasis through the methylation and downregulation of the *RUNX3* promoter [85]. Moreover, HOXA cluster antisense RNA 3 (HOXA-AS3) overexpression has been correlated with significantly shorter PFS and OS, whereas its downregulation has been found to inhibit the EMT process in vitro [68]. Interestingly, TC0101441 levels may be a potential independent prognostic marker of OS and DFS in EOC through the downregulation of *KISS1* (a metastasis suppressor gene) [108]. Furthermore, downregulation of AOC4P has been positively correlated with FIGO stage and LN metastasis by promoting EMT [123]. Increased PVT1 expression has also been associated with advanced FIGO stage and poor prognosis for OC patients. On the other hand, silencing of PVT1 impaired cell proliferation, migration, and invasion in vitro. More specifically, the EMT-related promotion of OC by PVT1 involved its interaction with *EZH2*, which resulted in the repression of miR-214 expression [70]. Accordingly, LINC01296 overexpression was negatively linked with progression-free survival (PFS) and OS, whereas knockdown of LINC01296 inhibited EMT in OC [97]; the underlying mechanism of its oncogenic function involves the regulation of EMT through the LINC01296/miR-29c-3p axis [98]. A negative correlation between HOXD-AS1 expression and PFS and OS in EOC patients has been also reported. Two EMT-associated mechanisms were identified in current literature. More specifically, this lncRNA acts as a ceRNA (i) via the miR-186-5p/PIK3R3 pathway [111] and (ii) via the miR-133a-3p/Wnt/β-catenin pathway [112]. lncARSR expression was also correlated with FIGO stage, histological grade, LN metastasis, and poorer OS. In terms of mechanism, lncARSR was found to increase *ZEB1* and *ZEB2* expression by sponging members of the miR-200 family [117]. Following the conduction of an integrated analysis, Mitra et al. identified 3 EMT-related lncRNA (*DNM3OS*, *MEG3* and *MIAT*) out of which *DNM3OS* was found to be associated with worse OS [120]. Another lncRNA, HOXA11, has been significantly associated with histological grade and preoperative CA-125, as well as EMT promotion and shorter PFS and OS, in patients with serous OC [121]. Accordingly, HOTAIR and CCAT1 expression were highly positively correlated with FIGO stage, histological grade, LN metastasis, and reduced OS [114,122]. The pro-metastatic effects of HOTAIR were partially mediated by the regulation of EMT-related genes [122] whereas CCAT1 promoted OC progression via a miR-152/miR-130b-ADAM17/WNT1/STAT3/*ZEB1* regulatory network [114]. Finally, CCAT1 was found to act as a ceRNA by directly sponging miR-490-3p and thus elevating TGFβR1 and promoting EMT [113]. 

On the other hand, various lncRNAs have been associated with the inhibition of EMT phenotype in OC (Figure 3). 

Table 2 summarizes the tumor-suppressing lncRNAs that have been implicated in OC-related EMT. The tumor-suppressing properties of MIR503HG were linked with (i) increased expression of mesenchymal markers and decreased expression of epithelial markers and (ii) decreased expression of the transmembrane protein with EGF-like and two follistatin-like domains 1 (TMEFF1). At the same time, MIR503HG knockdown resulted in TMEFF1 upregulation, which consequently activated the MAPK and PI3K/AKT pathways [124]. Wang et al. investigated the dynamics between Linc00261 and EMT in high-grade serous OC. The authors reported that Linc00261 acts as a negative regulator of OC-related EMT by targeting the miR-552/ATG10 axis [125]. Another lncRNA exerting tumor-suppressing properties in EOC is host gene 10 (SNHG10). Indeed, SNHG10 downregulation has been correlated with poor prognosis via the miR-200a-3p/BIN1 axis [126]. Furthermore, HCG11 also inhibited EMT by sponging miR-1270 to upregulate *PTEN* [127], whereas the lncRNA RP11-499E18.1 exhibited similar action by regulating the RP11-499E18.1/*PAK2*/*SOX2* axis [128]. In addition, LINC-PINT overexpression was reported to suppress cell proliferation, migration, invasion, and EMT while promoting apoptosis in OC, serving as a sponge to miR-374a-5p [129]. Accordingly, the tumor-suppressing properties of WDFY3-AS2 were the result of it sponging miR-18a and increasing RORA expression [130]. Zeng et al. reported that LIMT played a crucial role in inhibiting OC tumorigenesis regulated by EGFR signaling. Interestingly, the authors found that EGF secreted by M2-like TAMs inhibited LIMT expression in OC cells, facilitating EMT and tumor progression [131]. ADAMTS9-AS2 impeded cell proliferation, invasion, and EMT in OC cells both in vitro and in vivo. The underlying mechanism indicated that this lncRNA acted as a ceRNA for miR-182-5p and thus regulated FOXF2 expression [132]. 

To date, very few studies have investigated the role of EMT-related lncRNAs in OC drug resistance (Table 3). Indisputably, one of the major concerns in OC treatment relates to platinum resistance and there is evidence that EMT plays a critical role. EMT-regulated cisplatin resistance has been implicated with (a) the lncRNA CHRF/miR-10b axis via the activation of STAT3 signaling [133], (b) H19 overexpression [134] and (c) lncRNA HCP5 upregulation [135]. TMPO-AS1 silencing was found to inhibit EMT, invasion, migration and 5-FU resistance of OC cells via the miR-200c/TMEFF2 axis and the disruption of the PI3K/Akt signaling pathway [136].

## 5. EMT-Related lncRNAs: Current Evidence

Taken together, the above-mentioned studies indicate a direct or indirect association between lncRNA signatures and EMT-mediated OC progression, which could potentially lead to the discovery of novel biomarkers for this deadly disease. The retrieved lncRNAs from our review are further discussed in the following sections for their diagnostic, prognostic, and therapeutic potential.

### 5.1. EMT-Related lncRNAs in Cancer Diagnosis

Several EMT-related lncRNAs that were retrieved in this review have already been evaluated for their value as diagnostic biomarkers in various cancer types. One of the best-recognized lncRNAs with high diagnostic accuracy is MALAT1. As seen in Table 1, MALAT1 may possess diagnostic efficacy for OC [77]. Notably, it has also exhibited similar potential in non-small-cell lung cancer [area under the curve (AUC): 0.79] [137]. Moreover, it has been proposed as a biomarker for the early diagnosis of prostate cancer (PC); its expression was found to be positively correlated with high Gleason score and PSA level and it has been patented (CN104498495) as an auxiliary biomarker for early PC diagnosis, particularly for PSA “gray area” cases [138,139]. A meta-analysis by Chen et al. concluded that the differential expression of MALAT1 in various tumor types, including OC, can support its diagnostic potential [140]. Based on our review, NEAT1 has been proposed as a therapeutic target for OC [93]. However, this lncRNA has exhibited high diagnostic potential for NSCLC, displaying both high sensitivity and specificity [141]. Similarly, while HOTAIR has been associated with poor prognosis in OC [122], it also exhibits good characteristics as a diagnostic biomarker for multiple tumors, such as diffuse large B-cell lymphoma (DLBCL) (AUC: 0.71; sensitivity, 72.6%; specificity, 69.7%) [142], glioblastoma multiforme (GBM) (area under the ROC curve: 0.913; sensitivity, 86.1%; specificity, 87.5%) [143], and esophageal squamous cell carcinoma (ESCC) (AUC: 0.793) [144]. Notably, HOTAIR and MALAT1 are also under patent (CN105586399A) as diagnostic biomarkers for papillary thyroid cancer. Another relevant lncRNA with diagnostic potential is H19. More specifically, the AUC of plasma lncRNA H19 was found to be 0.81 in breast cancer patients (sensitivity, 56.7%; specificity, 86.7%), a finding which is higher than that of traditionally used breast cancer biomarkers such as carbohydrate antigen 153 (CA153) and carcinoembryonic antigen CEA [145]. Moreover, the diagnostic ability of H19 was high for gastric cancer (GC) (AUC: 0.838) [146] as well as papillary thyroid cancer [147]. The diagnostic and prognostic efficacy of exosomal HOTIPP has been also suggested for GC, displaying higher AUC compared to other biomarkers such as CEA, CA 19-9 and CA 72-4 [148]. Interestingly, the value of CCAT1 as a diagnostic biomarker is being currently investigated in a clinical trial (NCT04269746) in colorectal cancer (CRC). Indeed, the combination of CCAT1 and HOTAIR levels has been reported to display good diagnostic characteristics in CRC [149]. Finally, the additional evaluation of CCAT2 levels to CA-125 and squamous cell carcinoma antigen (SCC) levels was suggested to improve the diagnosis of cervical cancer [150].

### 5.2. EMT-Related lncRNAs in Cancer Prognosis

The aberrant expression of oncogenic lncRNAs correlates with unfavorable effects regarding OS, metastasis, and tumor stage and grade; thus lncRNAs could potentially serve as prognostic biomarkers. In the following paragraph, we will briefly discuss the prognostic value of EMT-related lncRNAs in multiple tumor types. In addition to OC [151], MALAT1 has been also proposed as a prognostic marker for CRC [152], hepatocellular carcinoma (HCC) [153], squamous cell lung cancer [154], BC [155,156], GC [157], esophageal cancer [158], and osteosarcoma [159]. Moreover, MALAT1 and HOTAIR levels have been suggested as a putative predictive duo of neuroendocrine transformation in PC [160]. The expression signature of HOTAIR has been also associated with poor prognosis in various cancer types, such as bladder cancer [161], GC [162], and CRC [163]. According to our review, CCAT1 has been proposed as an EMT-related lncRNA with putative prognostic value in OC. Similarly, its expression has been correlated with aggressive malignant phenotypes in HCC [164], lung cancer [165], and cholangiocarcinoma [166]. Accordingly, H19 prognostic value has been assessed in acute myeloid leukemia, CRC, and melanoma [167,168,169]. Finally, beyond its diagnostic value, HOTTIP displays additional potential as a prognostic biomarker. Indeed, its expression has been associated with poor prognosis in multiple tumor types [170,171,172,173,174].

### 5.3. EMT-Related lncRNAs as Therapeutic Targets in Cancer

The therapeutic potential of EMT-related lncRNAs represents another very promising scenario. Their abnormal expression and important roles in crucial cellular processes make them putative targets for cancer therapy. For example, BC-819, a double stranded DNA plasmid including the promoter H19 and the gene for diphtheria toxin A, has been investigated in clinical trials [175,176]. Mechanistically, H19 facilitates the expression of diphtheria toxin in cancer tissues, thus inhibiting tumor growth [177]. Of note, a phase 1/2a clinical trial evaluated the optimal dose, preliminary efficacy, and safety of BC-819 following intraperitoneal administration in patients with recurrent ovarian/peritoneal cancer. According to the results, this agent exhibits a good safety profile, but further research investigating the combination of intraperitoneal chemotherapy and BC-819 should be assessed in phase 2 and 3 trials [176]. Although some other lncRNAs are being investigated for their therapeutic potential [177,178,179,180], such treatment applications in the everyday clinical setting still have a long way to go.

## 6. LncRNAs as Biomarkers and Therapeutic Targets: Future Perspectives and Challenges

In recent years, circulating lnc RNAs have provided new insights into the mechanism of oncogenesis and opened up new biomarker possibilities. In terms of their diagnostic and prognostic potential, lncRNAs exhibit several advantages. Interestingly, it has been suggested that they have higher sensitivity and specificity compared to traditional markers [181,182]. In addition, they can be easily detected in bodily fluids (such as blood, saliva, and urine) by using common molecular biology techniques (i.e., qRT-PCR, microarray hybridization, and RNA sequencing), serving as non-invasive and low-cost biomarkers. More importantly, lncRNAs can be relatively stable in bodily fluids, escaping from degradation by ribonucleases and DNases [183], and finally, their dysregulation throughout treatment can offer a valuable monitoring tool for disease status and drug resistance. LncRNAs have been also proposed as therapeutic targets for several pathologic conditions, including cancer. This fact is predominantly based on the discovery that oncogenic lncRNAs can enter the bloodstream by loading into exosomes and then be transported to distant non-cancerous target cells, hence favoring their transformation to cancer cells [184]. On this basis, tumor-derived exosomes appear to offer a window of opportunity for novel anticancer therapies by serving as a vehicle for transferring selective cargos, such as lncRNA mimics with tumor-suppressing properties or inhibitors of oncogenic lncRNAs, aiming to suppress the formation of secondary sites.

However, before establishing the usefulness of lncRNAs as biomarkers, it is critical to address several issues. First, it is important to contemplate pre-analytical errors that may result in ambiguous and biased measurements. Pre-analytical variables that should be taken under consideration include the origin (e.g., whole blood, serum, plasma), processing (e.g., volume, type of anticoagulant), and storage conditions (e.g., temperature) of the biological sample [185]. Another topic concerns technical limitations regarding the methods used for RNA isolation. Various extraction and quantification techniques have been employed so far, each of which displays advantages and disadvantages. For example, in terms of extraction, column-based protocols have been suggested to be more reliable compared to guanidine/phenol/chloroform-based methods due to possible organic and phenolic contamination of the RNA extract when using the latter technique. In addition, the NanoDrop spectrophotometer, a frequently used instrument for circulating RNA quantification, has questionable sensitivity in the case of plasma/serum samples [186]. Another challenge relates to the credible measurement of circulating lncRNAs. Despite the implementation of several molecular techniques such as qRT-PCR, RNA-seq, and lncRNA microarray platforms [187], there are still various issues that should be addressed (such as cost per sample, specific equipment, data analysis, and normalization) before they can be reliable and feasible for day-to-day laboratory analysis and clinical practice. Therefore, developing sensitive and specific detection, quantification, and standardization methods for lncRNAs is crucial for their successful translation to clinically relevant biomarkers. In addition to these technical issues, other variables with the potential to interfere with the prompt interpretation of results are associated with inter-patient variability and environmental factors. Apart from demographic variables (such as race, age, and gender), which can be relatively simple to analyze, lncRNA single nucleotide polymorphisms (SNPs) may also affect lncRNA expression among patients [188]. Moreover, other donor-related factors, such as diet and physical activity, have been shown to contribute to changes in ncRNA levels, and hence the measured concentrations may reflect a synopsis of the individual state rather than the status of a disease [189,190]. Moreover, intra-patient variability may also influence lncRNA levels, given that several treatments may also affect their circulating levels over time.

To date, the majority of published research regarding lncRNA dysregulation has been conducted using whole plasma or urine samples. However, in the past decade, extracellular vesicles (EVs), and particularly exosomal ncRNAs, have become a research hotspot among investigators. Indeed, the differential expression of exosomal ncRNAs in samples from cancer patients has been reported in a plethora of studies. However, despite continuous improvements in EV isolation and characterization techniques, several questions remain regarding the accurate extraction and quantification of exosomal ncRNAs. For example, other molecular complexes may be also present in the vesicles when using many commercially available kits. On the other hand, ultracentrifugation may be more accurate in terms of exosome isolation but is time-consuming and thus unattractive for biomarker discovery. Finally, although exosomes are thought to be relatively stable [191], the need to establish univocal methodologies is important for generating reproducible and reliable results.

In view of the observations mentioned above, future endeavors should focus on the conduct of randomized trials in order to validate lncRNA signatures (single or panel) that have been reported in the current literature. Although the clinical investigation of lncRNAs has already moved to the level of clinical trials, there are still several parameters that should be considered when designing such studies so that a well-founded association between the differential expression of lncRNAs and disease pathology is ensured. Theoretically, an ideal investigation should include (i) an adequate sample size (in order to minimize selection bias) and (ii) cancer-specific lncRNA profiles with high specificity and sensitivity, pre-defined variations (regarding inter-/intra-individual differences and environmental factors), precise kinetics, and proper normalization with endogenous controls in bodily fluids.

Overall, a thorough co-evaluation of the variables that may affect the measurement of lncRNAs is critical for developing and implementing standard operating procedures for their analysis and application as reliable biomarkers. Therefore, the standardization of methodologies, normalization of procedures, and uniformization of sample processing is imperative to reduce interlaboratory variability and improve data interpretation. Furthermore, a comprehensive understanding of the crosstalk between the molecular functions of lncRNAs and cancer pathology is crucial to comprehend their differential expression and ultimately generate a consensus in validating lncRNAs as clinical biomarkers.

## 7. Concluding Remarks

The need to for sensitive and specific identification of novel OC biomarkers is more than clear. LncRNAs display a great deal of promise and clinical significance in cancer. Their high specificity in cancer cells constitutes a unique characteristic that makes them effective candidates as novel and efficient biomarkers. Moreover, the detection of differentially expressed lncRNAs in bodily fluids indicates that they may serve as non-invasive and low-cost biomarkers. However, despite that current literature includes a plethora of studies investigating their potential roles, the actual mechanisms by which they function remain not fully understood. Indeed, the expression patterns of lncRNAs in OC and how they interact with other factors involved in OC progression need to be further addressed in future studies in order to identify new efficient biomarkers and design effective therapeutic approaches to OC. In this context, various lncRNAs that regulate OC-related EMT possess high potential to serve as novel diagnostic or prognostic markers and could be targeted to restrict OC progression.

## Figures and Tables

**Figure 1 ijms-24-10079-f001:**
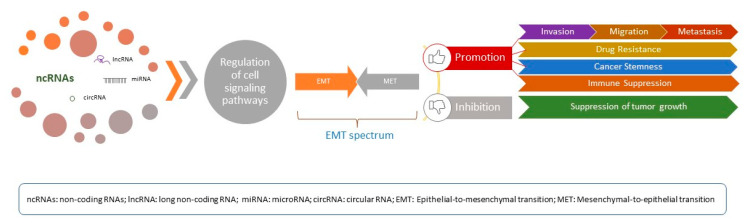
The crosstalk between non-coding RNAs and EMT. NcRNAs can act as oncogenes or tumor suppressors, regulating genes that modulate cell cycle-associated pathways and thus promoting or inhibiting the EMT process.

**Figure 2 ijms-24-10079-f002:**
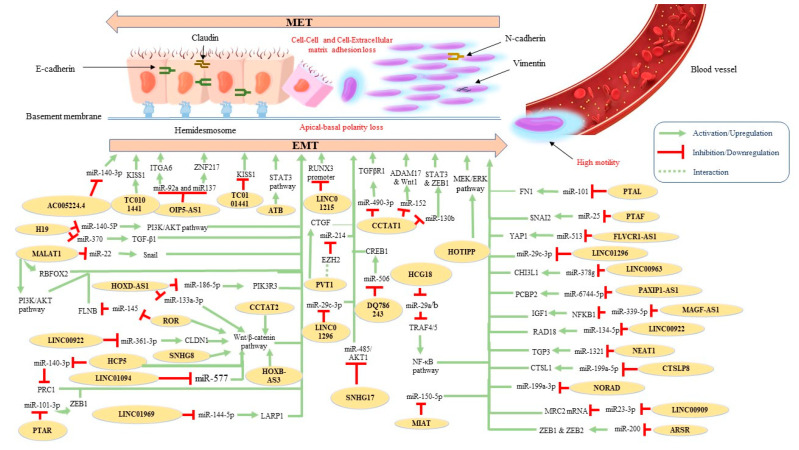
Proposed molecular mechanisms of identified oncogenic EMT-related lncRNAs in ovarian cancer.

**Figure 3 ijms-24-10079-f003:**
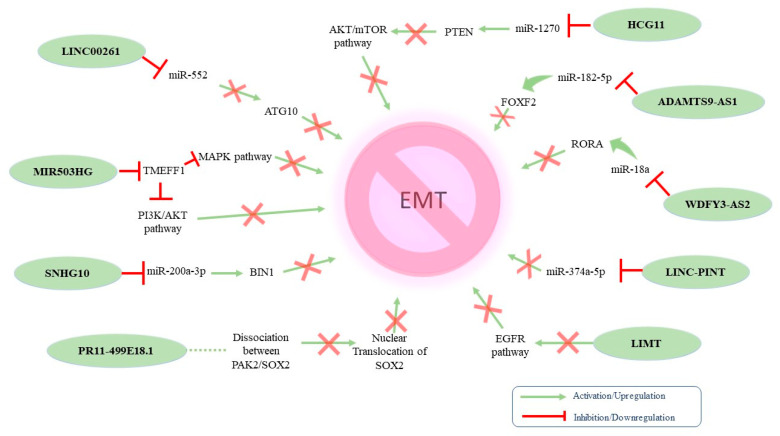
Proposed molecular mechanisms of identified tumor-suppressing EMT-related lncRNAs in ovarian cancer.

**Table 1 ijms-24-10079-t001:** List of oncogenic lncRNAs implicated with ovarian cancer-related EMT.

lncRNA	Expression	Result	Expression of EMT Markers/Genes	Potential Application	Subtype of OC	Ref.
AC005224.4	↑	EMT promotion in vitro	AC005224.4 overexpression resulted in decreased E-cadherin expression and increased expressions of N-cadherin, *Snail*, and vimentin, whereas its knockdown led to the opposite effects	Therapeutic target	-	[67]
HOXA-AS3	↑	EMT promotion in vitro	HOXA-AS3 knockdown resulted in E-cadherin upregulation and β-catenin, AKT, and vimentin downregulation	Prognostic marker and therapeutic target	EOC	[68]
PVT1	↑	EMT promotion in vitro	PVT1 knockdown suppressed CTGF and vimentin expression but led to E-cadherin overexpression	Therapeutic target	-	[69]
		EMT promotion	PVT1 knockdown resulted in (i) the downregulation of vimentin, β-catenin, transcription factors, *Snail*, and *Slug* and (ii) E-cadherin upregulation	Diagnostic marker and therapeutic target	EOC	[70]
lncRNA SNHG17	↑	EMT promotion	SNHG17 knockdown upregulated E-cadherin level and reduced the levels of N-cadherin and vimentin	Therapeutic target	EOC	[71]
OIP5-AS1	↑	EMT promotion	OIP5-AS1 inhibition increased E-cadherin and decreased vimentin expression	Therapeutic target	-	[72]
			OIP5-AS1 knockdown increased E-cadherin levels and decreased N-cadherin, vimentin, *Slug*, and *Twist* levels	Therapeutic target	EOC	[73]
HCG18	↑	EMT promotion	HCG18 knockdown increased E-cadherin levels, decreased MMP2, MMP9, and vimentin, and downregulated *ZEB1*, *Slug*, *TWIST1*, and *Snail*	Therapeutic target	EOC	[74]
MIR210HG	↑ (under hypoxia conditions)	EMT promotion	MIR210HG knockdown increased E-cadherin and decreased N-cadherin expression under hypoxic conditions	Therapeutic target	-	[75]
MALAT1	↑	EMT promotion	Upregulation of MALAT 1 resulted in increased levels of SNAIL protein	Therapeutic target	-	[76]
				Diagnostic marker/therapeutic target	EOC	[77]
			MALAT1-regulated RBFOX2 plays a pro-tumorigenic role in OC, and one of its potential targets, *KIF1B*, may also affect tumorigenicity	-	EOC	[78]
			-	Therapeutic target	EOC	[79]
lncBCAS1-4_1	↑	EMT promotion	lncBCAS1-4_1 upregulation led to significant increase of N-cadherin and vimentin as well as EMT-related transcriptional factor (*ZEB1*)	Therapeutic target	-	[80]
SRA	↑	EMT promotion	SRA knockdown increased the levels of E-cadherin and decreased the expression of β-catenin, N-cadherin, *Snail*, and vimentin; SRA overexpression triggered the opposite effects	Predictive biomarker	-	[81]
LINC00909	↑	EMT promotion	Ectopic LINC00909 expression decreased E-cadherin and increased N-cadherin/vimentin levels; LINC00909 knockdown partially reversed the process of EMT	Diagnostic biomarker	-	[82]
LINC00922	↑	EMT promotion	LINC00922 knockdown led to E-cadherin overexpression and inhibits the expression of vimentin	Therapeutic target	-	[83]
PAXIP1-AS1	↑	EMT promotion	PAXIP1-AS1 knockdown led to decreased levels of MMP2, MMP9, and N-cadherin and increased E-cadherin expression	-	-	[84]
LINC01215	↑	EMT promotion	LINC01215 knockdown led to decreased expressions of MMP-2, MMP-9, and vimentin and increased the expression of E-cadherin	Therapeutic target	EOC	[85]
MAFG-AS1	↑	EMT promotion	MAFG-AS1 led to increased E-cadherin expression and reduced N-cadherin, MMP-2, MMP-9, and vimentin expression	Therapeutic target	-	[86]
CTSLP8	↑	EMT promotion	CTSLP8 knockdown increased E-cadherin expression and decreased N-cadherin expression. In addition, overexpression of CTSLP8 upregulated, while CTSLP8 knockout suppressed the expression of *ZEB1* and *Snail*	Therapeutic target	-	[87]
LINC01969	↑	EMT promotion	LINC01969 knockdown increased E-cadherin expression but decreased *Snail* and vimentin expression levels	Prognostic biomarker	-	[88]
E2F4as	↑	EMT promotion	E2F4as knockdown resulted in increased expression of E-cadherin and decreased expression of N-cadherin, β-catenin, vimentin, Wnt-5β, *Snail*, and claudin-1 expression	Diagnostic, prognostic biomarker and therapeutic target	-	[89]
LINC01094	↑	EMT promotion	LINC01094 knockdown upregulated E-cadherin expression and downregulated vimentin expression	Therapeutic target	-	[90]
HOTTIP	↑	EMT promotion	HOTTIP knockdown led to elevated E-cadherin and reduced N-cadherin, vimentin, and *Snail* expression	Therapeutic target	-	[91]
DSCR8	↑	EMT promotion	DSCR8 overexpression decreased E-cadherin expression and increased N-cadherin and vimentin expression	Therapeutic target	-	[92]
NEAT1	↑	EMT promotion	NEAT1 knockdown led to E-cadherin upregulation and N-cadherin and vimentin downregulation	Therapeutic target	-	[93]
LINC00858	↑	EMT promotion	LINC00858 knockdown led to E-cadherin activation and N-cadherin, Slug, and Twist protein inhibition	-	-	[94]
SNHG8	↑	EMT promotion	SNHG8 knockdown led to increased expression of E-cadherin and decreased levels of N-cadherin and *Snail*	-	-	[95]
HCP5	↑	EMT promotion	HCP5 silencing led to E-cadherin upregulation and vimentin downregulation	Therapeutic potential	-	[96]
LINC01296	↑	EMT promotion	LINC01296 knock down led to increased E-cadherin expression and decreased expression of N-cadherin and vimentin	Diagnostic and prognostic biomarker and therapeutic target	-	[97]
						[98]
MIAT	↑	EMT promotion	MIAT knockdown led to increased expression of E-cadherin and decreased expression of N-cadherin, *Snail*, and *ZEB1*	Prognostic biomarker/therapeutic target	-	[99]
NORAD	↑	EMT promotion	NORAD knockdown led to increased levels of E-cadherin and decreased N-cadherin and vimentin levels, whereas overexpression of NORAD, triggered opposite effects	Therapeutic target	-	[100]
LINC00963	↑	EMT promotion	LINC00963 knockdown increased E-cadherin and decreased vimentin levels	Therapeutic target	-	[101]
ATB	↑	EMT promotion	LncRNA-ATB knockdown decreased the expression of p-STAT3 and vimentin and increased E-cadherin expression	Therapeutic target	-	[102]
H19	↑	EMT promotion	H19 knockdown increased E-cadherin levels and decreased the levels of *Snail* and vimentin, whereas overexpression of H19 had the opposite effect	Therapeutic target	-	[103]
H19	↑	EMT promotion	H19 overexpression led to increased E-cadherin and vimentin and decreased N-cadherin	Therapeutic target	-	[104]
AC004988.1 (PTAL)	↑ in mesenchymal subtype samples	EMT promotion	PTAL silencing resulted in E-cadherin and ZO-1 upregulation and N-cadherin, vimentin, and *Slug* downregulation	Therapeutic target	serous OC	[105]
ROR	↑	EMT promotion	ROR knockdown led to increased E-cadherin expression and decreased N-cadherin and vimentin expression	Therapeutic target	-	[106]
ROR	↑	EMT promotion	ROR silencing led to higher E-cadherin expression and lower vimentin, β-catenin, and c-myc levels	Therapeutic target	HGSOC	[107]
TC0101441	↑	EMT promotion	TC0101441 knockdown increased the expression of E-cadherin but decreased the expression of N-cadherin and *Snail*	Prognostic marker and Therapeutic target	EOC	[108]
FLVCR1-AS1	↑	EMT promotion	FLVCR1-AS1 knockdown upregulated E-cadherin expression and downregulated vimentin and *Snail* expression	Therapeutic target	serous OC	[109]
HOXB-AS3	↑	EMT promotion	HOXB-AS3 knockdown led to elevation of E-cadherin levels and repressed N-cadherin and vimentin levels	Prognostic biomarker and Therapeutic target	EOC	[110]
HOXD-AS1	↑	EMT promotion	HOXD-AS1 downregulation, upregulated E-cadherin expression and downregulated vimentin expression	Therapeutic target	EOC	[111,112]
CCAT1	↑	EMT promotion	CCAT1 knockdown enhanced E-cadherin and claudin expression and reduced vimentin, N-cadherin, and MMP9 expression	Prognostic biomarker and therapeutic target	-	[113]
			CCAT1 downregulation promoted E-cadherin expression and reduced vimentin and N-cadherin expression, while CCAT1 upregulation had the opposite results	Diagnostic and prognostic biomarker and therapeutic target	EOC	[114]
DQ786243	↑	EMT promotion	HOXA-AS3 knockdown resulted in E-cadherin upregulation and β-catenin, AKT, and vimentin downregulation in OC cell lines	Therapeutic target	-	[115]
PTAR (see erratum)	↑	EMT promotion	PTAR downregulation led to increased E-cadherin expression and reduced fibronectin1, *ZEB1*, and vimentin expression	Therapeutic target	serous OC	[116]
ARSR	↑	EMT promotion	Overexpression of lncARSR reduced E-cadherin and ZO-1 and increased N-cadherin and vimentin	Therapeutic target	EOC	[117]
PTAF	↑	EMT promotion	PTAF knockdown led to increased E-cadherin expression and decreased *SNAI2* expression	Therapeutic target	serous OC	[118]
CCAT2	↑	EMT promotion	CCAT2 knockdown led to upregulation of E-cadherin and downregulation of N-cadherin, *SNAI*, and *Twist*	Therapeutic target	EOC	[119]
DNM3OS, MEG3, and MIAT	↑	EMT promotion	DNM3OS knockdown led to elevated of E-cadherin and reduced levels of N-cadherin, *Snail*, and *Slug*	Therapeutic target	HGSOC	[120]
HOXA11	↑	EMT promotion	HOXA11 knockdown led to E-cadherin upregulation and N-cadherin, β-catenin, and vimentin downregulation. Moreover, the expression of *Twist* and *Snail* were also downregulated	Prognostic biomarker and Therapeutic target	serous OC	[121]
HOTAIR	↑	EMT promotion	HOTAIR knockdown resulted in increased expression of E-cadherin and decreased vimentin and *Snail* expression	Prognostic biomarker and Therapeutic target	EOC	[122]

EMT: epithelial-to-mesenchymal transition; HGSOC: high-grade serous ovarian cancer; EOC: epithelial ovarian cancer; OC: ovarian cancer; CTGF: connective tissue growth factor; MMP: matrix metalloproteinase; *ZO*-*1*: Zonula occludens-1; ↑: increased.

**Table 2 ijms-24-10079-t002:** List of tumor-suppressing lncRNAs implicated with ovarian cancer-related EMT.

lncRNA	Expression	Result	Expression of EMT Markers/Genes	Potential Application	Subtype of OC	Ref.
MIR503HG	↓	EMT inhibition	MIR503HG knockdown suppressed E-cadherin expression and increased N-cadherin and vimentin expression	Therapeutic target	-	[124]
Linc00261	↓	EMT inhibition	Linc00261 knockdown led to E-cadherin downregulation and increased expression of *Slug*, *Twist1*, and N-cadherin, whereas the opposite was observed following Linc00261 overexpression	Therapeutic target	HGSOC	[125]
SNHG10	↓	EMT inhibition	SNHG10 overexpression led to increased E-cadherin levels and decreased N-cadherin, vimentin, and *Snail* levels	Predictive biomarker and therapeutic target	EOC	[126]
HCG11	↓	EMT inhibition	Ectopic expression of HCG11 increased E-cadherin levels and reduced N-cadherin expression	Therapeutic target	-	[127]
RP11-499E18.1	↓	EMT inhibition	RP11-499E18.1 overexpression led to increased E-cadherin and decreased vimentin expression. RP11-499E18.1 knockdown exerted the opposite effects	Diagnostic marker	-	[128]
LINC-PINT	↓	EMT inhibition	LINC-PINT silencing was associated with decreased levels of E-cadherin and high N-cadherin and vimentin levels	Target for OC treatment	-	[129]
AOC4P	↓	EMT inhibition	MMP9 and COL1A2 genes were upregulated in two AOC4P-siRNA-transfected cell lines	Target for anti-metastatic strategies	EOC	[123]
WDFY3-AS2	↓	EMT inhibition	WDFY3 overexpression increased E-cadherin levels and decreased N-cadherin and vimentin levels	Therapeutic potential	-	[130]
LIMT	↓	EMT inhibition	Co-culturing of OC cells with M2-like TAMs (which suppress LIMT) downregulated the expression of E-cadherin while N-cadherin and vimentin were upregulated	Diagnostic, prognostic and therapeutic potential	EOC	[131]
ADAMTS9-AS2	↓	EMT inhibition	ADAMTS9-AS2 overexpression induced E-cadherin expression and decreased vimentin expression while ADAMTS9-AS2 inhibition resulted in the opposite effects	Therapeutic target	-	[132]

EMT: epithelial-to-mesenchymal transition; HGSOC: high-grade serous ovarian cancer; EOC: epithelial ovarian cancer; OC: ovarian cancer; TAMs: tumor-associated macrophages; MMP: matrix metalloproteinase; siRNA: small interfering RNA; ↓: decreased.

**Table 3 ijms-24-10079-t003:** List of EMT-related lncRNAs in ovarian cancer drug resistance.

lncRNA	Expression	Result	Expression of EMT Markers/Genes	Potential Application	Subtype of OC	Ref.
HCP5	↑	EMT phenotype in CR cells	-	Prognostic value	-	[135]
CHRF	↑	EMT promotion in CR cells	CHRF downregulation led to decreased levels of E-cadherin and increased vimentin levels in CR cells	Therapeutic target for sensitizing CR cells	-	[133]
TMPO-AS1	↑	EMT promotion	TMPO-AS1 silencing led to increased levels of E-cadherin and decreased levels of vimentin	Therapeutic potential	-	[136]
H19	↑	EMT promotion	*Twist*, *Slug*, and *Snail* were dramatically upregulated and E-cadherin decreased in CR cells	Therapeutic target for sensitizing CR cells	-	[134]

EMT: Epithelial-to-mesenchymal transition; CR: cisplatin resistant; OC: ovarian cancer; ↑: increased.

## Data Availability

Not applicable.

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
