# Peer review of "The Role of EMT-Related lncRNAs in Ovarian Cancer"

_ijms, 2023, doi:10.3390/ijms241210079_

Round 1
Reviewer 1 Report
In this manuscript, Lampropoulou et al. focused on LncRNAs, and did a review on the role of EMT-related LncRNAs in ovarian cancer. Based on the overall quality of the manuscript, it is not suitable to be accepted as it is for now. Please find the reasons below.
1. The content in the section of Discussion is basically the same with the main body of this manuscript. Moreover, usually it is not necessary to include a part of “discussion” in review articles.
2. The manuscript is not logically and clearly arranged.
3. The manuscript aims to focus on the potential of EMT-related LncRNAs as markers for diagnosis and prognosis prediction and targets for the therapy of ovarian cancer. However, not much was mentioned about the necessary properties as good markers and targets. Not sufficient review was made from different aspects regarding this topic.
4. The manuscript is not well written, with a lot of typographical and grammar mistakes. The whole flow is really confused and hard to follow.
5. In line 18, it is not necessary to include the word “type” here.
6. In line 19, “in OC” could be removed from here.
7. In line 124, it should be “importance of EMT”.
8. In section 4.1, it is not very clear how it was divided into several paragraphs.
9. In line 428, review articles usually give systemic review on the development and current status of a certain topic, but is not able to “identify” original findings.
The manuscript is not well written, with a lot of typographical and grammar mistakes.
Reviewer 2 Report
In my opinion, the analyzed topic is interesting enough to attract the readers’ attention. The goal of this article was to discuss the role of lncRNAs in regulating OC-related EMT and their underlying mechanisms. I think that the abstract of this article is well organized and clear.
In my opinion, the discussion could be studied in depth and extended. Maybe, it could be useful the evaluation of what is known about ovarian cancer pathogenesis. Furthermore it could be interesting to assess the preoperative predictors of the status of these kind of patients. In particular I suggest these two articles in ordrt to get deeper in the topic: PMID: 25987014 and The role of preoperative frailty assessment in patients affected by gynecological cancer: a narrative review Ottavia D’Oria, Tullio Golia D’Auge, Ermelinda Baiocco, Cristina Vincenzoni, Emanuela Mancini, Valentina Bruno, Benito Chiofalo, Rosanna Mancari, Riccardo Vizza, Giuseppe Cutillo, Andrea Giannini Vol. 34 (No. 2) 2022 June, 76-83 doi: 10.36129/jog.2022.34 . Because of these reasons, the article should be revised and completed. Figures and tables are clear. Considered all these points, I think it could be of interest for the readers and, in my opinion, it deserves the priority to be published after major revisions.
minor editing is needed
Round 2
Reviewer 1 Report
The manuscript has been sufficiently improved.
Author Response
Dear Reviewer 1,
Thank you again for your time and valuable input.
Sincerely,
Christos Papadimitriou
Professor of Therapeutics-Oncology
National and Kapodistrian University of Athens
Athens,11527,
Greece
Reviewer 2 Report
I read with great interest the Manuscript titled "The Role of EMT-related lncRNAs in Ovarian Cancer”, topic interesting enough to attract readers' attention.
The quality of the manuscript has improved thanks to the changes made. The methodology is accurate, and conclusions are supported by data analysis. References are relevant to the search
Considering all these points, I think it could be of interest to the readers and, in my opinion, it deserves the priority to be published.
Author Response
Dear Reviewer 2,
Thank you again for your valuable time and input.
Sincerely,
Christos Papadimitriou
Professor of Therapeutics-Oncology
National and Kapodistrian University of Athens
Athens,11527,
Greece